# Fluorescence Labeling of Cellulose Nanocrystals—A Facile and Green Synthesis Route

**DOI:** 10.3390/polym14091820

**Published:** 2022-04-29

**Authors:** Lorenzo Donato Campora, Christoph Metzger, Stephan Dähnhardt-Pfeiffer, Roland Drexel, Florian Meier, Siegfried Fürtauer

**Affiliations:** 1Department of Civil and Industrial Engineering, University of Pisa, 56122 Pisa, Italy; l.campora@studenti.unipi.it; 2Fraunhofer Institute for Process Engineering and Packaging IVV, 85354 Freising, Germany; 3Chair of Process Systems Engineering, TUM School of Life Sciences Weihenstephan, Technical University of Munich, 85354 Freising, Germany; christoph.metzger@tum.de; 4Microscopy Services Dähnhardt GmbH, 24220 Flintbek, Germany; spfeiffer@microscopy-consulting.com; 5Postnova Analytics GmbH, 86899 Landsberg am Lech, Germany; roland.drexel@postnova.com (R.D.); florian.meier@postnova.com (F.M.)

**Keywords:** cellulose nanocrystals, silanization, fluorescence labeling, green chemistry

## Abstract

Efficient chemical modification of cellulose nanocrystals (CNCs) by grafting commonly involves aprotic solvents, toxic reactants, harsh reaction conditions, or catalysts, which have negative effects on the particle character, reduced dispersibility and requires further purification, if products are intended for biomedical applications. This work, in contrast, presents a robust, facile, and green synthesis protocol for the grafting of an amino-reactive fluorophore like fluorescein isothiocyanate (FITC) on aqueous CNCs, combining and modifying existent approaches in a two-step procedure. Comparably high grafting yields were achieved, which were confirmed by thermogravimetry, FTIR, and photometry. The dispersive properties were confirmed by DLS, AF4-MALS, and TEM studies. The presented route is highly suitable for the introduction of silane-bound organic groups and offers a versatile platform for further modification routes of cellulose-based substrates.

## 1. Introduction

The polysaccharide cellulose is important for humankind ever since due to its natural abundance, versatile applicability, and renewable character. Traditionally, cellulose is utilized in forest products and used to produce paper and cardboard [1], natural and synthetic textile fibers [2] (e.g., cotton and viscose), cellulose-derived and cellulose reinforced biopolymers [3,4,5], and many more [6]. Cellulose nanocrystals (CNCs), extracted from natural cellulose, have the potential to play a major role in the 21st century in the development of advanced materials. CNCs constitute a unique and renewable building block, on which materials with improved performance and new functionality can be built [7,8,9,10]. They are already used, or prospected, as emulsion stabilizer, sustainable catalyst, templating agent for mesoporous nanostructures, and for various energy and electronic system applications [11]. Their outstanding properties qualify CNCs for further future technologies; for instance, as reinforcement material (the specific elastic modulus of CNCs is potentially stronger than steel and similar to Kevlar^®^ [7]) and as barrier coating (CNCs’ oxygen permeability is comparable to fossil-based coating materials [12,13]). Furthermore, it can form highly porous sponges for various applications, such as adsorption of contaminants and oil spillages from water [14,15,16,17,18,19,20,21,22].

The exposed hydroxy groups on the CNC surface can be utilized as starting points for various modification approaches via grafting [7], aiming at increased hydrophobicity, improved compatibility with matrix polymers, better dispersibility, or conjugation of small molecules. Due to its excellent biocompatibility, biodegradability, and low toxicity, biomedicine research recently focused on CNCs as the basis for controlled delivery systems [23], anti-microbial materials [24,25], specific molecules probes [26,27], and tissue engineering substrates [28]. By introducing fluorescent molecules to the nanoparticles’ surfaces, CNCs can be functionalized with fluorescent labeling ability. This promotes the use in biomedical fields [29], such as optical bio-imaging, as biosensor [30,31,32,33,34], and photodynamic therapy [35]. Furthermore, labeling of CNC with fluorescing molecules allows nanoparticle characterization by various fluorescence techniques, easier tracing for toxicity, and bioactivity studies from materials into living cells [36,37,38,39,40,41,42,43,44,45,46,47,48,49].

Various fluorescent labels were reported to be covalently attached to the surface of CNCs [36]. These comprise fluorescent molecules with an effective leaving group, such as succinimidyl esters of fluorescein and oregon green [37], succinimidyl ester of 1-pyrenebutyric acid [38], rhodamine B isothiocyanate (RBITC) [39], bromopyrene dyes [40], supramolecular and azido-terpyridine-ligands [41], PEI-chlorin derivatives [42], 5-(4, 6-dichlorotriazinyl)-aminofluorescein [43], and 7-amino-4-methylcoumarin [44], but also unspecific binding molecules such as Calcofluor White [45,46,47]. Using approaches from synthetic peptide chemistry, surface grafting of fluorescing amino acid molecules offers biologically active building blocks on CNCs [50,51]. Furthermore, combined approaches with (i) an initial activation of the CNC surface by grafting and (ii) further conjugation of fluorescent molecules have been reported ((i) Fmoc-Gly and (ii) peptide bonding with 7-methylcoumarin-oligopeptide [48]; (i) Fmoc-L-leucine and (ii) thiourea formation with 5(6)-carboxy-2’,7’-dichlorofluorescein (CDCF) [49]; (i) thioglycolic acid and (ii) thiol-ene click reaction of quinidine [52]).

In this work we developed an approach to conjugate fluorescein isothiocyanate (FITC) to amino-modified CNCs in aqueous solution, which has some advantages compared to previously reported synthesis strategies (detailed discussion see Section 3.1). Those advantages refer to most of the twelve principles for green chemistry, which were postulated by Anastas and Warner [53], in particular prevention of waste, atom economy, less hazardous reactions, safe solvents, design of safer chemicals, energy efficiency, reduction of derivatives (e.g., protection groups), catalysis, biodegradation, green analytical chemistry, accident prevention, and use of renewable feedstocks.

Moreover, the presented route is a mild, aqueous, simple, and fast modification technique, is highly suitable for the introduction of an anchor group onto the CNCs’ surfaces, completes the chemical toolbox for CNC modification, e.g., conjugation with isocyanates, aldehydes, or amino acids [54], and provides further possibilities for the synthesis of novel functionalized cellulose derivatives.

## 2. Materials and Methods

### 2.1. Materials

CNC 6 wt.% in water or spray dried (type NCV100, (*c*(sulfate) = 246–261 mmol kg^−1^—manufacturer information [55]) was purchased from CelluForce (Montreal, Canada). 3-Aminopropyltriethoxysilan (APTES), fluorescein-5(6)-isothiocyanate (FITC, racemic mixture), HCl, NaOH, Na_2_CO_3_, NaHCO_3_, NH_4_Cl, DMSO and acetone were purchased from Sigma-Aldrich (Darmstadt, Germany).

### 2.2. Methods

#### 2.2.1. Titration Experiments

Aqueous solutions of CNC and APTES were titrated to understand their pH-shifts and therefore being able to optimize the following silanization protocols. A total of 100 mL of solution (1 wt.% in water each) were stirred vigorously with a magnetic stirrer before and during the titration experiments. The pH was continuously recorded with a calibrated FiveEasy pH-meter (Mettler Toledo, Gießen, Germany) with temperature correction. Either HCl or NaOH *c* = 0.1 mol L^−1^, respectively, was stepwise added to CNC 1 wt.% with a glass burette, changing the pH by 0.1–max. 1 unit (close to inflection point), recording the pH if reached a stable value. Aqueous solution of 1 wt.% APTES was titrated in a similar manner, but only with HCl *c* = 0.5 mol L^−1^. Molar ratios were plotted against pH and inflection points determined (discussion see Section 3.2.1 and Appendix A).

#### 2.2.2. Preparation of CNC-APTES

In a typical synthesis, 100 mL aqueous CNC suspensions with 1 wt.% were prepared from 6 wt.% CelluForce NCV100 by dilution with deionized water and stirring for 30 min at 1000 rpm on a magnetic stirring plate (C-Mag HS 10 digital, IKA, Staufen, Germany) to achieve a uniform dispersion. Reactions were conducted in 250 mL low-density polyethylene (LDPE) bottles (Nalgene™, Thermofisher Scientific Inc., Schwerte, Germany). First, the pH was lowered to 1.5 by addition of 4.5 mL 1 mol L^−1^ HCl, followed by addition of 0.004 moles (0.9 mL) APTES. Due to the alkaline nature of APTES, the pH raised to about 2.5. The samples were then continuously stirred for 30 min to allow the complete hydrolysis of the alkoxysilane to silanols (see Figure 1, step 1, left part of the reaction scheme). To induce the covalent bonding between the surface hydroxy groups of the CNCs and the silanols, the pH was adjusted to 10.0–10.5 by adding 1 mol L^−1^ NaOH to the mixture, which was then left under constant stirring for 3 h (see Figure 1, step 1, right part of the reaction scheme). The condensation reaction was generally performed at room temperature (air conditioning at 23 °C), but for one sample CNC-APTES-4 at increased temperature of 40 °C. For isolating the CNC-APTES from unreacted silanols, the CNC-APTES gels were centrifuged at 7000 rcf, washed with 20–30 mL water/acetone or acetone, and redispersed in an ultrasonic bath (Sonorex digitec DT 514, Bandelin, Berlin, Germany) for 5 min between each washing step. The gel recovered after the last centrifugation was used as starting material for the grafting reaction with FITC.

#### 2.2.3. Preparation of CNC-APTES-FITC

The APTES-modified CNCs (gel) were diluted with 0.1 mol L^−1^ sodium bicarbonate buffer (pH = 9) until the solid content was 1 wt.% (calculation based on dry matter obtained from drying at 60 °C overnight). A solution of 1 g L^−1^ FITC in DMSO was freshly prepared and stored in a dark container to avoid direct exposure to sunlight. About 12.5 mL of the FITC/DMSO solution were slowly added to 50 mL of the CNC-APTES and stirred overnight under exclusion of light and at 4 °C to allow the electrophilic addition of the isothiocyanate group to the primary amine group (see Figure 1, step 2). A total of 167 mg NH_4_Cl were dissolved in the mixture (resulting in a concentration of 50 mmol L^−1^) and stirred for two more hours to quench the reaction and block the non-reacted excess FITC. Washing was performed similar to the previously described process, but here 20 mL bicarbonate buffer was used for each washing step (up to six washing steps). Supernatants were kept for further analysis via photometry. The obtained gel had a bright yellow color and it remained stable after the washing steps.

#### 2.2.4. Thermogravimetry

Aqueous gels were dried overnight in a convection furnace (FD-S Solid.Line, Binder, Tuttlingen, Germany) at 60 °C until constant mass. Combustion residues of furnace-dried samples were determined by a TGA 701 S4C instrument (LECO, St. Joseph, MI, USA) via a quasi-static thermogravimetric program (isothermal steps at 105, 550 and 950 °C in O_2_ atmosphere). Samples with minimum masses of 0.25 g were weighed in ceramic crucibles and placed in a carousel for simultaneously analyzing up to 19 samples. During combustion, the samples are stepwise rotated and intermittently weighed on an integrated balance (accuracy 0.0001 g), until mass constancy at step temperature is reached.

#### 2.2.5. Calculation of Degree of Substitution (DoS)

The DoS is the molar ratio of grafted molecules, APTES, on cellulose (calculated as anhydroglucose unit, AGU). Two kinds of DoS are considered: the total degree of substitution (DoS_total_), related to the bulk material, and the degree of substitution on the surface (DoS_surface_), taking into account that only the surface of the CNC particles can be modified, but not each polymeric cellulose chain inside of the nanoparticles.

The DoS_total_ of APTES on CNCs was calculated from the obtained percentage of inorganic residue after combustion of CNC-APTES at 950 °C (see Section 2.2.4), since it can be directly related to present silane concentration in the grafted samples. It is assumed that the grafted silane mainly oxidizes to form the thermodynamic stable combustion product SiO_2_. Furthermore, the surface-bound ions (sodium hydrogen sulphate derivatives) on untreated CNCs, which are present because of the industrial production process, must be considered (combustion residue of untreated CNCs = 1.44 wt.%). Taking also into account that grafting of APTES onto CNCs could form up to three different covalent species (since each silanol has three accessible hydroxy groups), either R-Si(-OH)_2_-O-AGU, R-Si(-OH)(-O-AGU)_2_ or R-Si(-O-AGU)_3_ (R = aminopropyl-residue), three fitting functions for calculating the DoS_total_ from the combustion residue were formulated (see Appendix A). The average number of unbound hydroxy groups per silanol is negligible for the calculation of the DoS_total_, since its influence is lower than the magnitude of experimental error (DoS_total_ ± 0.0002–0.0004, depending on DoS_total_ of sample).

The DoS_surface_ is calculated based on the DoS_total_, considering a crude model of a nanocrystal to estimate the number of available sites for reaction on the surface [44]. Since particle diameters vary typically between 2.30 and 4.50 nm (square cross section) and they have lengths of 44.00–108.00 nm (manufacturer information [55]), we consider an average diameter of 3.40 nm and an average length of 76.00 nm. The glucan chain is approximately 0.57 nm wide, so there are 3.40/0.57 or 5.96 (~6) chains in each side, or about 35.58 (~36) chains in a nanocrystal cross-section. Lengthwise, a 76.00 nm long nanocrystal would have 76.00/0.57 = 133.33 AGUs. This makes a total of 35.58 × 133.33 = 4744 AGUs per nanocrystal. The surface of the nanocrystal, which is ideally considered as a cuboid, therefore exposes 4 × 5.96 × 133.33 + 2 × 5.96 × 5.96 AGUs = 3250 AGUs. The ratio of AGUs in bulk to AGUs on the surface gives the factor to calculate DoS_surface_ (1):DoS_surface_ = DoS_total_ × (4744/3250)(1)

#### 2.2.6. FTIR Analysis

Fourier transform infrared (FTIR) spectra were recorded with a Frontier^TM^ FTIR spectrometer L1280034, using the software PerkinElmer Spectrum IR, Version 10.6.1, both supplied by Perkin Elmer (Shelton, CT, USA). Measurements were performed at ambient temperature and ambient pressure on dried samples (higher signal intensity due to reduced water signal). An ATR (attenuated total reflectance/Golden Gate™ ATR System, type GS10500-Z, Specac Ltd., Orpington, UK) device with a diamond crystal top plate was utilized. A constant contact pressure was realized by tightening the securing screw with a ratchet, which is limited to a torque of 0.3 Nm. Scans were recorded within a wavenumber range of 4000 to 600 cm^−1^ at a resolution of 1 cm^−1^. Specific IR-bands were plotted with Origin 2018b (OriginLab Corporation, Northampton, MA, USA) and then compared.

#### 2.2.7. Photometry

The Uvikon XL UV/VIS spectrophotometer (BioTek, Bad Friedrichshall, Germany), equipped with a tungsten-halogen-lamp, was used for quantification of free and bound FITC molecules at an adjusted step-width of 1 nm within the wavelength range of 200–800 nm. Full spectra were recorded for all samples. Typical absorbance values were selected at 495 nm, which is the maximum absorbance of FITC. Standards of FITC in different solvents (water, acetone or bicarbonate buffer) were prepared in the linear range of 0.1–10 µmol L^−1^. Supernatants from washing steps were diluted to match concentrations within the linear range and were measured in standard cuvettes (aqueous samples in polystyrene (PS) cuvettes, acetone containing samples in quartz glass cuvettes). Furthermore, suspensions of FITC-functionalized CNCs were coated on glass slides (sodium glass, 1 mm diameter, Marienfeld, Lauda-Königshofen, Germany) and dried at 60 °C to produce semi-transparent films. Those films were measured in transmission via the use of an Ulbricht sphere. Obtained spectra were used for a qualitative proof of functionalization.

#### 2.2.8. Dynamic Light Scattering (DLS)

Samples were diluted to 0.025 wt.% with deionized water and then ultrasonicated in an ultrasonic bath for 60 min with intermittent vortexing. The suspensions were centrifuged (1000 rcf/5 min) to precipitate non-dispersed solids. The supernatant was filtered through a hydrophilic glass fiber membrane with a pore size of 1 μm (CHROMAFIL Xtra GF, Thermofisher Scientific Inc., Schwerte, Germany). CNC dispersions were equilibrated for 30 min at 25 °C in disposable polystyrene cuvettes before measuring the hydrodynamic apparent particle size of dispersed CNCs via a Zetasizer Nano ZSP (Malvern Instruments, Worcestershire, UK) equipped with a red laser (633 nm) under a backscatter detection angle of 173°. The z-average (*d*_h_) and the dispersity (*Ð*) from the cumulants analysis were obtained according to ISO 22,412 [56], using the available literature values for refractive index (1.468) and absorption (0.1) of cellulose. Each sample was measured in triplicate; *d*_h_ and *Ð* are reported as means with standard deviations.

#### 2.2.9. Asymmetrical Flow Field-Flow Fractionation—Multi-Angle Light Scattering (AF4-MALS)

Size-based fractionation of colloidal CNCs was performed at 25 °C using an AF4 system (AF2000) equipped with an autosampler (PN5300) and a channel thermostat (PN4020) (all provided by Postnova Analytics GmbH, Landsberg am Lech, Germany). The measurement setup and parameters for CNCs fractionation were reported elsewhere recently [57]. Additionally, the applied fractionation method is reported in the Appendix A. In brief, the CNC suspensions were diluted to 0.04 wt.%, filtered with 1 μm syringe filters, and then ultrasonicated for 45 min. Sample volumes of 20 μL were injected using 1 mmol L^−1^ NaCl as the eluent. Evaluations of CNC suspensions separated with AF4 showed high repeatability and reproducibility in accordance with the guidelines of ISO/TS 21,362 [58].

The scattered light intensities of fractionated CNCs were detected under 19 or 18 active angles, respectively, ranging from 12° to 156° with an on-line coupled MALS detector PN3621 (Postnova Analytics GmbH, Landsberg am Lech, Germany) at a wavelength of 532 nm and a cell temperature of 35 °C. A detailed description of the fit functions and calculations were also reported elsewhere recently [57]. The hydrodynamic radius of CNCs (*r*_h_) as a function of retention time was evaluated using FFF theory. Therefore, the effective channel height was determined from the retention time of fractionated polystyrene beads (Appendix A). Calculations were performed within the NovaAnalysis software (version 2007, Postnova Analytics GmbH, Landsberg am Lech, Germany) [59].

#### 2.2.10. Transmission Electron Microscopy (TEM)

Pioloform-coated copper grids (G2440C) from Plano (Wetzlar, Germany) were incubated with 0.1% poly-L-lysine at 23 °C for 30 min, rinsed with water, and dried under dust-free atmosphere. Never-dried CNC suspension was diluted with H_2_O to a concentration of 0.025 wt.% and homogenized in a low-intensity ultrasonic bath for 10 min before droplets of the suspension were applied to the grids with a pipette. After 20 min, the grids were rinsed with water and left for drying. Negative staining of CNCs was performed with a saturated ethanolic uranyl acetate solution. Excess liquid was removed, and the samples were dried again. Transmission electron microscopy (TEM) images were acquired with a Philips CM10 instrument, coupled with a CCD camera (IDS, Obersulm, Germany), at an acceleration voltage of 80 kV. The particle lengths, *l*_p_, were determined with ImageJ software on a particle number of *n*~100.

## 3. Results and Discussion

### 3.1. Overview and Discussion of Existing Routes for Cellulose Labelling with FITC

One of the most employed and widely available fluorescence dyes is fluorescein isothiocyanate (FITC). Various synthesis routes with different substrates are shown in Table 1. The direct reaction of FITC with the carbohydrate hydroxy groups produces low yield [37,60], which is expressed in a low DoS, and involves undesired catalysts such as dibutyltin dilaurate (DBTL), or solvents such as dimethylformamide (DMF) [61]. Therefore, primary amino groups are favorably introduced, which are more reactive toward FITC [54,62]. A well-described route is the etherification of the carbohydrate’s hydroxy group with epichlorohydrine (ECH), subsequently followed by a reaction of the epoxy-ring with ammonium hydroxide to form the primary amine [39,63,64,65], followed by thiourea formation with FITC.

Alternatively, the use of amino-bearing silanes, e.g., (3-aminopropyl)-triethoxysilane (APTES) or (3-aminopropyl)- trimethoxysilane (APTMS), requires less synthesis steps and circumvents the use of toxic ECH. Moreover, attaching fluorescein to nanoparticles by a spacer linker with 2–10 carbon atoms can effectively control the distance between the fluorescein and nanoparticles, and, therefore, prevent fluorescence quenching [66]. Much research has been performed on amino-silane modification of carbohydrates with or without subsequent FITC-labeling; not only for CNCs ([38,67,68]), but also for cellulose nanofibers (CNFs) [69], filter paper [70], starch nanocrystals (SNCs) [71], and bacterial nanocellulose (BNC) [72].

However, all these described synthesis routes implicate some disadvantages, for example the use of toxic solvents such as DMF [38], or the irreversible aggregation of CNCs by heating and drying (leading to “hornification”) [68]. Synthesis routes in aqueous solution using “never-dried CNCs”, which also follow the rules of green chemistry [53,73], overcome these shortcomings and are therefore urgently desired.

Such an approach, based on CNFs as substrate (green principles “renewable feedstocks” and “biodegradability”), is reported in the works of Hettegger et al. [74] and Beaumont et al. [75], where organosilanes were efficiently grafted by a pH-dependent two-step procedure in aqueous solution, while agglomeration is prevented (green principles “safe solvents”, “prevention of waste” and “atom economy”). The organo-triethoxysilanes (TES) bearing azidopropyl- (AzPTES), vinyl- (VTES), and mercaptopropyl-residues (MPTES) [72,76,77] were first hydrolyzed at low pH, followed by condensation at the CNFs’ surfaces at higher pH with a catalytic amount of base (green principle “less hazardous reactions” and “catalysis”).

Based on these studies, we designed a grafting protocol of APTES on CNCs, while maintaining the particle character of the CNCs. The so introduced amino groups were used for a coupling reaction with FITC (see Figure 1). For CNCs having a different surface chemistry to CNFs due to present sulfate esters, to the best of our knowledge a pH-controlled grafting protocol is not yet reported. Furthermore, APTES was hitherto not mentioned for pH-dependent grafting onto CNCs.

**Table 1 polymers-14-01820-t001:** Overview of similar functionalization strategies for FITC on carbohydrate substrates; (1) and (2) are the respective synthesis steps.

Substrate	Grafted Molecule (1)	Reaction Conditions (1)	Fluorophore (2)	Reaction Conditions (2)	Comment	DoS	Ref.
CNC	-	-	FITC, RBITC	dark reaction/0.1 mol L^−1^ NaOH/72 h		0.031	[37]
CNC	ECH	NaOH/60 °C, amination with NH_4_OH/60 °C	FITC	dark reaction/borate buffer/overnight	ECH is toxic	0.024	[63]
CNC	ECH		FITC, RBITC			0.024	[39]
CNC, CNF	ECH		FITC				[65]
CNC	ECH *	ECH + NH_4_OH first form 2-hydroxy-3-chloro propylamine; DMSO/TBAH/50 °C	-	-		0.357	[64]
CNC	APTES	DMF/2 h RT	FITC	DMF/19 h RT	DMF is toxic		[38]
CNC	APTES	modification with PDDA, Fe_3_O_4_; TEOS/APTES at alkaline pH	FITC	ethanol, 24 h/RT	superparamagnetic core-shell structure		[67]
CNC	APTES	hydrolysis of APTES at pH 4, + CNC, stirring 2 h/RT, precipitate cured at 105 °C	-	-	thermal curing	0.180	[68]
CNF	APTES	water/ethanol, pH 5.5, 1 h, RT; curing 110 °C	-	-		0.458	[69]
CNF	AzPTES, VTES, MPTES	acidic silane hydrolysis (HCl, 30 min, RT), alkaline condensation (NaOH, 3 h, RT)	-	-	aqueous protocol, pH induced hydrolysis	0.195	[75]
CNC	APTES		FITC	bicarbonate buffer, pH 9, overnight		0.040	this work

Abbreviations in this table: Dimethyl sulfoxide (DMSO), tetrabutylammonium hydroxide (TBAH), poly (diallyldimethylammonium chloride) (PDDA), tetraethyl orthosilicate (TEOS); room temperature (RT). * Synthesis in which ECH is aminated before being grafted.

### 3.2. Preparation and Optimization of Aqueous Protocol

#### 3.2.1. Influence of Substrate and Silane Type on pH

The silanization protocol from Beaumont et al. [75] using CNFs as substrate, which was the starting point of this study, had to be modified with respect to the different surface chemistry of the here used CNCs and grafting agent APTES. The previously used silanes [75] involve thiol-, azide-, and vinyl-residues, which only have minor effect on the final pH of the mixture—this is in contrast with primary amine-containing APTES, which is of alkaline nature. Furthermore, in contrast to CNFs [75], CNCs expose sulfate ester groups at the particles surface, therefore acting as a heterogeneous buffer system. Most probably the sulfate groups do not take part in the reaction with APTES as educts, but have an influence on the zeta potential of the particles (ζ = −37 mV; manufacturer information [55]). They could have a catalytic effect on the hydrolysis of APTES due to their acidic nature, and furthermore reduce the grafting yield due to steric effects (blocking of available reaction sites).

To understand how the pH of the mixture is affected by different molar amounts of HCl (pre-hydrolysis of APTES) and NaOH (condensation reaction), titrations of the starting materials were performed (see Appendix A). The inflection point of the APTES titration was at an equimolar ratio with the acid at pH 4.5. The pK_b_ for protonation of the primary amine was found to be at 4.7, which is close to the literature value of 3.63 [78]. Below pH 6.4 the cloudy mixture of APTES in water turned transparent and clear, which indicates hydrolysis into the silanol. However, the interpretation of the CNC titration is less straightforward, since there are two inflection points at pH 4.5 and 8.0, probably occurring from differing surface chemistries. Below pH 4.5, the sulfate esters of the CNCs are considered to be fully protonated, forming the respective acid CNC-O-SO_3_-H. The pK_a1_ is between 2 and 3, which would be close to the pK_a_ of free HSO_4_^−^ (pK_a_ = 1.96 [79]). The upper pK_a2_ is at 6.5, which is in the region of HSO_3_^−^ (pK_a_ = 7.20 [79]). Those mixed species are likely to be present at the CNCs’ surfaces, formed during the hydrolysis with sulfuric acid during production (compare also Figure 1: only the deprotonated sulfate ester is shown).

By strictly using the same stoichiometric amounts of CNCs, HCl, and silane as in literature [75] for the hydrolysis step, the pH of the mixture was 10.4, followed by 1 mmol NaOH for condensation (pH = 10.8). The high pH throughout both reaction steps promoted self-condensation of APTES [80,81], which was indicated by a white precipitate of silica after centrifugation. Such precipitates from self-condensation of silanes at high pH values were also confirmed by literature [75], particularly observed when the reaction time was longer than 3 h. Therefore, for all further synthesis approaches, higher amounts of acid were used to achieve the low pH for the pre-hydrolysis of APTES (8–9-fold higher than in literature [75]). The progress of the pH value from the start of synthesis and during hydrolysis and condensation for different batches can be viewed in Appendix A.

#### 3.2.2. Increased Reaction Temperature

The synthesis protocol for silanization was also tested at increased temperature (40 °C, sample CNC-APTES-4) during a 3-h condensation reaction at alkaline conditions, whereas all other parameters were kept constant (see Section 2.2.2). However, the DoS increased only slightly (from 0.040 at 25 °C reaction temperature to 0.047 at 40 °C reaction temperature). Increased reaction temperature presumably increases grafting yield. However, since hornification of CNCs would occur at higher temperatures (50–60 °C), the applicability of temperature increase as controlling parameter is limited.

#### 3.2.3. Separation of CNC-APTES and CNC-APTES-FITC from Supernatants

After the condensation step of APTES onto CNCs (typically pH 10–10.5, 3 h), non-reacted APTES and residual ions from neutralization had to be removed from the solid CNC-APTES phase. In previous works, similar was done via dialysis [39,63,64,65], despite the need of time and consumption of water is quite high for that process. Desired time and resource saving centrifugation of CNC-APTES from non-aqueous solvents (such as DMF [38]), thermally cured CNC-APTES [68] or better precipitable APTES-modified cellulose nanofibers (CNF-APTES) [75] were reported to be successful. Thermally cured CNF-APTES could even be separated from the supernatant by vacuum filtration [69]. However, the washing and separation process of CNC-APTES from the aqueous phase via centrifugation had to be improved, since aqueous CNC dispersions have a high colloidal stability. The effectivity of separation in this study was visually assessed (homogenous, cloudy dispersion before and after centrifugation = no separation; clear phase separation into white solid and transparent clear supernatant = excellent separation; clear phase separation, but negligible turbidity of supernatant = good separation). Removal of the supernatant by centrifugation before the first washing step was feasible (7000 rcf, 5 min). If only deionized water was used for washing (slurrying and ultrasonication), the resulting dispersion could not be separated by centrifugation, even at higher relative centrifugal forces (15,000 rcf). Therefore, the chosen washing solution was a mixture of acetone/water for the first washing step, and acetone for the second washing step, both showing excellent separation performance already at 7000 rcf for 5 min.

The precipitate after the second washing step was used for further functionalization with FITC according to the described protocol in Section 2.2.3, resulting in CNC-APTES-FITC. Unbound FITC was removed like in the previous procedure, but with bicarbonate buffer as washing solution instead of acetone. The concentration of free FITC in the supernatant after centrifugation (7000 rcf, 5 min, good separation) was determined by photometry at 495 nm (0–10 µmol L^−1^, calibration curves see Appendix A). The ratio of FITC concentration in extracts from successive steps was 3.0–3.3, which reflects Nernst’s distribution law (see Appendix A). As can be seen from Figure 2, already after the third washing step there was no significant FITC concentration detectable in the supernatant (following steps extract <1 wt.% of total FITC). Although the concentration of FITC in further extracts would asymptotically approach a zero value, for practical reasons, three washing steps would be sufficient for effective FITC removal. The cumulative FITC content in the washing supernatants is 57 wt.% of the initial amount of FITC (=9.13 µmol of initial 16.05 µmol), therefore the differing 6.92 µmol are covalently grafted or physisorbed onto CNC-APTES.

### 3.3. Proof of Functionalization and DoS

Figure 3 shows the FTIR spectra of CNCs, CNC-APTES-3, and CNC-APTES-FITC-3, respectively. All samples clearly share similar spectral properties at wavelengths of 3340 cm^−1^ and 3272 cm^−1^ (intermolecular O-H bonding), 1648 cm^−1^ (adsorbed water molecules), 2895 cm^−1^, 1429 cm^−1^, 1370 cm^−1^, and 1310 cm^−1^ (C-H related modes), 1055 cm^−1^ (C-O), 1030 cm^−1^, 1105 cm^−1^ and 1162 cm^−1^ (C-O-C pyranose), 665 cm^−1^ (O-H out-of-plane), 895 cm^−1^ (β-glycosidic linkage), which are characteristic for cellulose-based materials [82,83,84]. Furthermore, all samples showed spectral peaks at 1200 cm^−1^ and 1335 cm^−1^, related to organic sulfate groups. For further details regarding the origin of specific IR-modes, see Appendix A. The magnified sections of the spectra in Figure 3 reveal that both CNC-APTES-3 and CNC-APTES-FITC-3 have peaks at 980 cm^−1^, related to amorphous silica (Si-O-Si) [85], and at 845–830 cm^−1^ related to Si-C bonds [75]. Both are a qualitative indication of a successful grafting procedure, although new covalent Si-O-C linkages between the silane and the CNC substrate cannot be determined by FTIR clearly. Furthermore, IR-modes solely visible in the spectrum of CNC-APTES-FITC-3 can be assigned to the covalently linked fluorophore (lactone 5-membered ring at 1775 cm^−1^ [86], C = O stretching mode at 1720 cm^−1^ and aromatic C-H out-of-plane bending at 875 cm^−1^ [82]).

Furthermore, photometric spectra of CNCs and CNC-APTES-FITC-3 were recorded in transmission (dry films on glass plates) as shown in Appendix A. The significant peak of the spectrum of CNC-APTES-FITC-3 in the region between 450 and 530 nm has a maximum at 504 nm, which is slightly higher than the photometric absorption maximum of molecular FITC at 495 nm. The formation of the thioester presumably modulates the excitation energy of the entire dye molecule, leading to lower excitation energies, related to higher absorption wavelengths.

The DoS_total_ of APTES on CNCs is 0.040–0.047 (CNC-APTES-3 and CNC-APTES-4, respectively); see Table 2. Grafting at higher pH levels (CNC-APTES-1 and CNC-APTES-2) results in virtually higher DoS, probably due to precipitation of APTES. Exemplarily for CNC-APTES-3, the DoS_surface_ according to Equation (1) (see Section 2.2.5) would be 0.058, which means that almost every 17th superficial AGU is grafted with a silane molecule, or every 50th exposed hydroxy group is bound.

For the calculation of the DoS of CNC-APTES regarding FITC (DoS_total_(FITC)) there are following considerations: For CNC-APTES-FITC-3 it was found that 6.92 µmol FITC are covalently bound to 0.5 g CNC, since consecutive washing steps could not remove this amount. The molar amount of 0.5 g CNC is 3083.37 µmol (AGU), therefore according to Equation (2)
DoS_total_(FITC) = 6.92 µmol/3083.37 µmol = 0.002 (2)

Inserting in Equation (1), the DoS_surface_(FITC) is 0.003. Hence, the molar ratio between grafted FITC and grafted APTES is 0.003/0.058, meaning that in sample CNC-APTES-3 there are 5.6 mol% of primary amine groups grafted with FITC.

### 3.4. Particle Size Characterization after Grafting

Aqueous suspensions of modified and non-modified CNCs were analyzed by dynamic light scattering (DLS) and asymmetrical flow field-flow fractionation (AF4) hyphenated to a multi-angle light scattering (MALS) detector (methods see Section 2.2.8 and Section 2.2.9). Cumulants analysis of DLS gives the intensity weighted hydrodynamic particle diameter, *d*_h_, and the polydispersity index, *Ð*, which are shown in Table 3 (mean and standard deviation of triplicates). DLS results revealed that the size distribution of modified CNCs is only slightly increased compared to non-modified CNCs (10–16.5%), while dispersity decreases (8–34%).

To investigate the polydispersity and potential changes in the size distribution of the modified CNCs, size-based fractionation and analysis by AF4-MALS were applied. In contrast to DLS, which is prone to an overestimation of larger size fractions [87] due to their stronger scattering signals, the monodisperse size fractions after AF4 fractionation can be investigated by light scattering without interference of larger particles. The MALS data evaluation yields the radius of gyration (*r*_g_). Moreover, the elution time was correlated to the corresponding hydrodynamic radius (*r*_h_) of the fractionated analytes using FFF theory [88]. AF4-MALS interestingly showed higher *r*_g_ and *r*_h_ values for those samples grafted at 25 °C (CNC-APTES-3 and CNC-APTES-FITC-3), whereas the sample synthesized at 40 °C had a similar size distribution as non-modified CNCs (see Table 4 and Figure 4a,b). From the fractogram and *r*_h_ size distribution the width of the distributions is directly accessible. For sample 3, the size distributions, for both *r*_g_ and *r*_h_, were broader and included larger size fractions. Furthermore, the ratio *r*_g_/*r*_h_ (shape factor) indicated a comparable shape of the non-modified CNCs and all modified CNC samples. The different reaction temperature for grafting of APTES on CNCs for samples 3 and 4 has obviously an influence on the dispersive properties of the CNCs, meaning less agglomeration for grafting at 40 °C. However, self-condensation of APTES could also be promoted at higher temperatures, resulting in less extensive silanization of the CNC surface, and, therefore, less interaction of the CNCs.

Clear evidence for the particle character of CNCs was provided by visualization with TEM. Discussed samples CNC, CNC-APTES-3, CNC-APTES-FITC-3, and CNC-APTES-FITC-4 (see Figure 5a–d) show distinguishable particles with clear outlines and no significant agglomerates. From each sample, the particle lengths, *l*_p_, were determined on a particle number of *n* ~100 (Figure 6—bar chart). The Gaussian size distribution (Figure 6—line plot) reveals that the particle lengths are not significantly affected by the treatment (see also means and standard deviations, Table 5), in a range of approx. ± 5%.

## 4. Conclusions

Labeling of synthetic molecules, like organic fluorophores, on biomaterials such as CNCs could be an experimental challenge. Commercial labelling kits, although powerful tools, are often limited to substrates like proteins, therefore not directly applicable and usually very costly. The toolbox of organic chemistry offers reactions, which need aprotic solvents, uses harsh temperatures and potentially toxic catalysts. These conditions could conflict the stability of CNC suspensions or lead to undesired contaminations and by-products, which pose an obstacle if labeled CNCs are intended e.g., for biomedical purposes.

Therefore, a robust, facile, and green synthesis protocol for the grafting of an amino-reactive fluorophore like FITC onto CNCs was developed, combining and modifying existent approaches in a two-step procedure. The advantage of the presented protocol is the use of never-dried aqueous CNCs (no solvent exchange necessary), the abandonment of toxic educts (for potential higher biocompatibility), and the simplicity of the approach (low cost and easy to reproduce). The grafting yield of APTES onto CNCs was comparably high with a DoS of 0.040 (bulk)/0.058 (surface), which was proven by thermogravimetry. The product CNC-APTES-FITC had a bright yellow color and has its absorption maximum around 504 nm. The release of unbound FITC during washing was monitored by photometry, the cumulative concentrations were used to demonstrate that 5.8 mol% of the amino-groups were linked to a fluorophore. Further evidence of functionalization was given by specific Si-C-bands in the solid product. Various techniques were applied for the determination of the particle size before and after the grafting procedure, which is an indicator for potential agglomeration of CNCs. While TEM images (particle length *l*_p_) showed that morphological features of CNCs at different synthesis steps did not significantly change, DLS (*d*_h_) revealed slightly increased average size for one synthesis condition. Similarly, the presence of larger size fractions was also confirmed by AF4-MALS. The presented mild, aqueous, and simple modification technique, which did not significantly affect the particle character of the CNC substrate, is highly suitable for the introduction of a silane-bound organic group and offers a versatile platform for further modification routes of cellulose-based substrates.

## Figures and Tables

**Figure 1 polymers-14-01820-f001:**
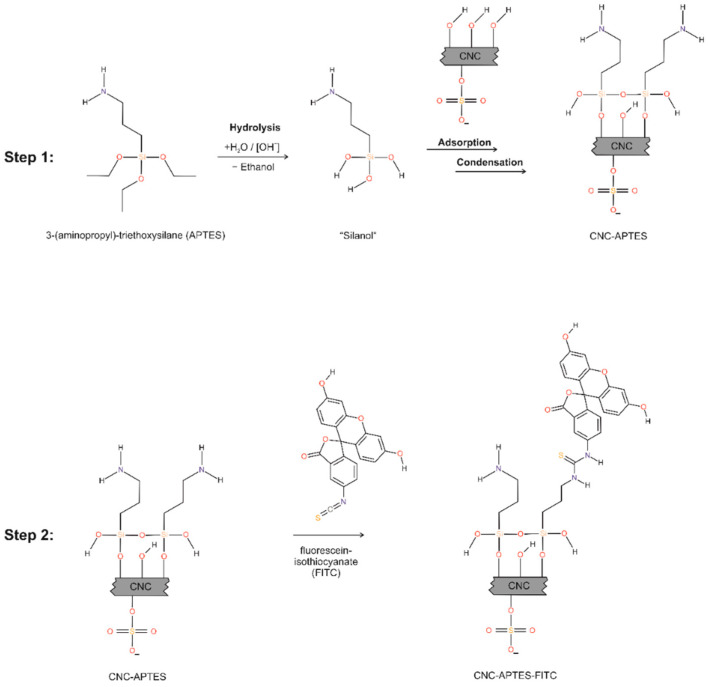
Reaction steps for synthesis of CNC-APTES and CNC-APTES-FITC.

**Figure 2 polymers-14-01820-f002:**
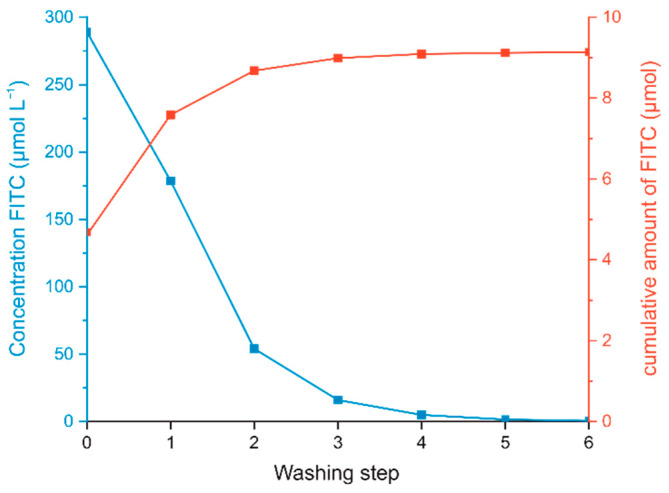
Stepwise washing of CNC-APTES-FITC-3 with bicarbonate buffer: FITC concentration in supernatant of each washing step (blue); removed molar amount of FITC from precipitate (cumulative, red).

**Figure 3 polymers-14-01820-f003:**
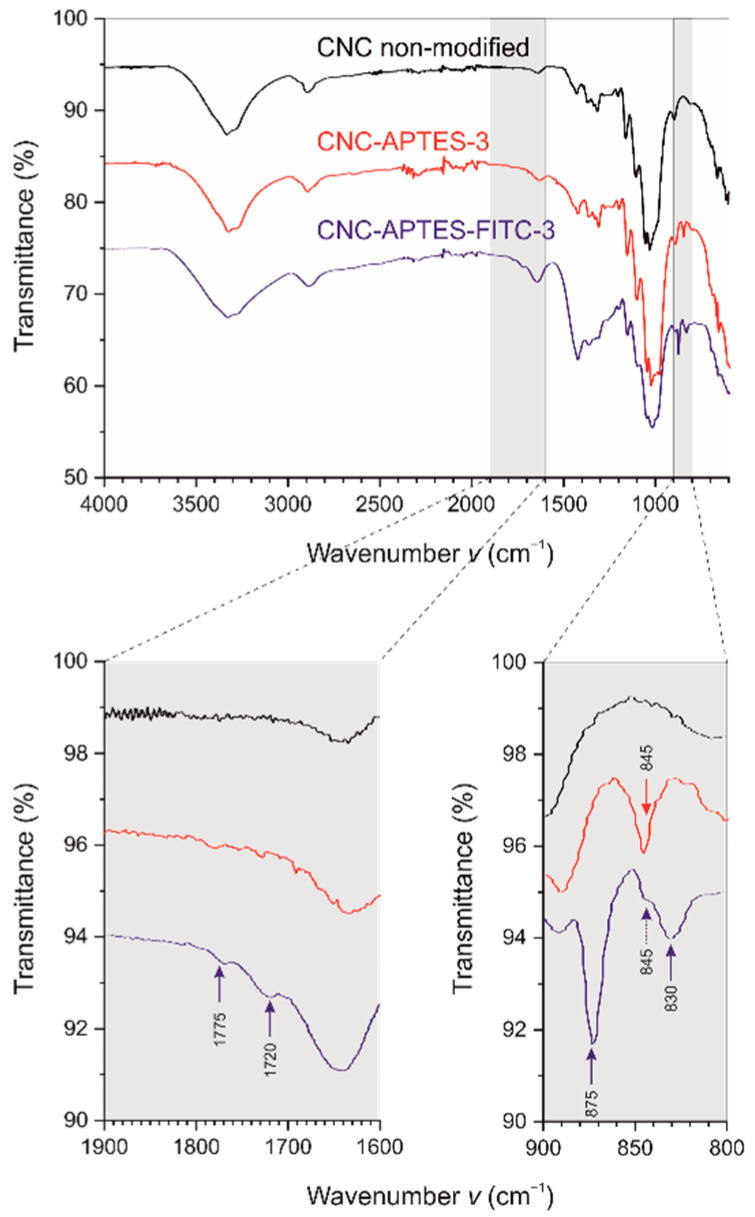
IR-spectra of CNC non-modified, CNC-APTES-3 and CNC-APTES-FITC-3.

**Figure 4 polymers-14-01820-f004:**
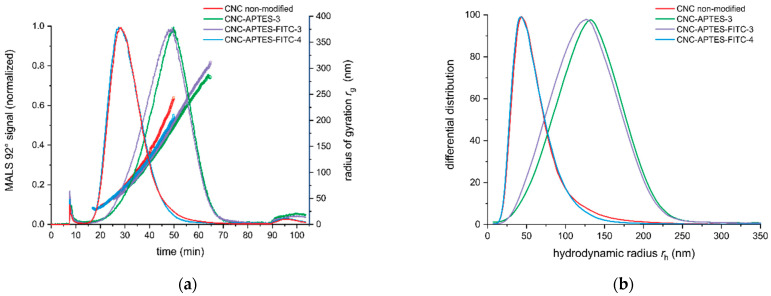
AF4-MALS (**a**) elution profile: MALS 92° signal (line plot) and radius of gyration *r*_g_ (scatter plot) (**b**) *r*_h_ size distribution.

**Figure 5 polymers-14-01820-f005:**
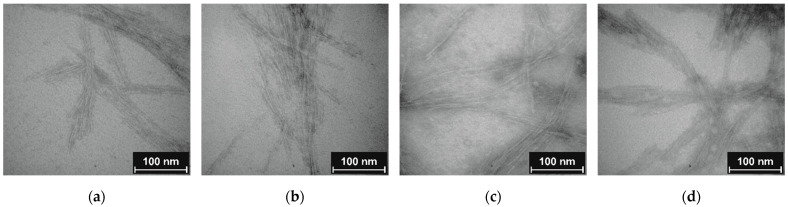
TEM image (**a**) non-modified CNC; (**b**) CNC-APTES-3; (**c**) CNC-APTES-FITC-3; (**d**) CNC-APTES-FITC-4.

**Figure 6 polymers-14-01820-f006:**
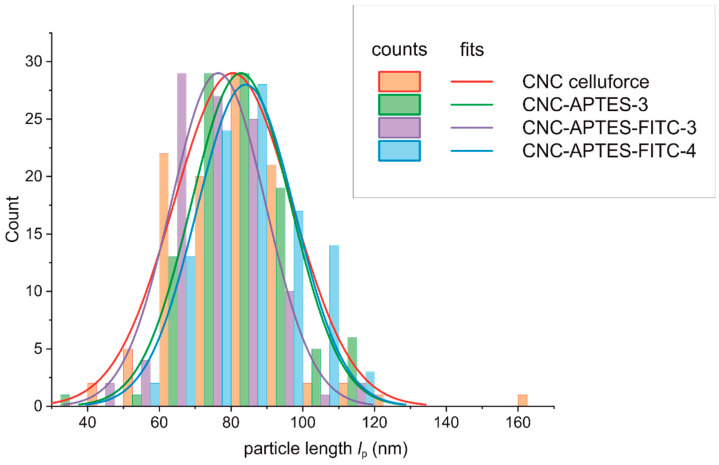
Particle size distribution from TEM imaging.

**Table 2 polymers-14-01820-t002:** DoS of APTES on CNCs for different synthesis protocols.

Sample	Preparation	Combustion Residue from Silane (wt.%)	DoS_total_	DoS_surface_
CNC-APTES-1	protocol according to [75]	3.68 * ± 0.63	0.106 * ± 0.020	0.155 * ± 0.030
CNC-APTES-2	pH establishment during hydrolysis	2.06 * ± 0.00	0.058 * ± 0.005	0.085 * ± 0.007
CNC-APTES-3	optimized protocol (25 °C)	1.42 ± 0.16	0.040 ± 0.005	0.058 ± 0.007
CNC-APTES-4	increased condensation temperature (40 °C)	1.69 ± 0.14	0.047 ± 0.004	0.069 ± 0.006

* Virtual high DoS due to precipitation of silane (high initial pH).

**Table 3 polymers-14-01820-t003:** *d*_h_ and *Ð* determined by DLS.

	DLS	
	*d*_h_ (nm)	*Ð*
CNC non-modified	163.4 ± 2.3	0.211 ± 0.005
CNC-APTES-3	190.4 ± 1.3	0.139 ± 0.011
CNC-APTES-FITC-3	182.8 ± 2.5	0.168 ± 0.012
CNC-APTES-FITC-4	178.3 ± 0.5	0.195 ± 0.018

**Table 4 polymers-14-01820-t004:** *r*_g_, *r*_h_ at the MALS 92° signal maximum obtained by AF4-MALS as well as the ratio *r*_g_/*r*_h_.

	AF4-MALS		
	*r*_g_ (nm)	*r*_h_ (nm)	*r*_g_/*r*_h_
CNC non-modified	63.5 ± 1.0	42.8 ± 2.2	1.49 ± 0.06
CNC-APTES-3	183.5 ± 1.5	133.7 ± 1.4	1.37 ± 0.02
CNC-APTES-FITC-3	179.8 ± 2.6	125.9 ± 3.3	1.43 ± 0.02
CNC-APTES-FITC-4	60.7 ± 1.4	41.8 ± 2.3	1.45 ± 0.09

**Table 5 polymers-14-01820-t005:** Length of particles *l*_p_, measured on TEM images (*n*: number of measured particles per sample).

	*l*_p_ (nm)	*n*
CNC non-modified	80.3 ± 16.7	105
CNC-APTES-3	82.7 ± 13.9	103
CNC-APTES-FITC-3	74.6 ± 13.2	100
CNC-APTES-FITC-4	84.1 ± 13.7	101

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
