# Peer review of "Fluorescence Labeling of Cellulose Nanocrystals—A Facile and Green Synthesis Route"

_polymers, 2022, doi:10.3390/polym14091820_

Round 1
Reviewer 1 Report
Dear authors article entitled Fluorescence labeling of cellulose nanocrystals – a facile and green synthesis route” describes an extended study on the modification of cellulose nanocrystals. The CNCs from many years are in area of interest to many scientists. Despite the visible effort and great contribution of the authors, the article requires attention in several places.
Comments;
- The introduction section is well written and includes all the necessary information. However, the modified schema 1 could be moved to material and methods (2.2.2. Preparation of CNC-APTES-FITC). The information’s collected in table 1 are rather usable for discussion of obtained results. I suggest removing table 1 from the introduction section.
- Authors wrote “By strictly using the same stoichiometric amounts of CNCs, HCl, and silane as in literature [63] (sample CNC-APTES-1: 100ml 1wt.% CNC = 6.17 mmol, 0.5 mmol HCl, 4mmol APTES”. In this section, the experimental details should be avoided authors should be focused on the discussion of the results.
- Authors determine the pK values for modified and raw CNC by simple titration. However, for CNC size determination authors used Zetasizer Nano ZSP. that device can be used also for Zeta potential determination which is a very important parameter when polymer surface is modified?
Moreover, any experimental details were not included in the material and methods section about pK determination?
- Authors mention in the result and discussion about testing higher temperatures for silanization of CNC. However, in materials and methods, any information was not included. Why only 40oC was tested as an alternative temperature?
- CNF-APTES the abbreviation of CNF needs an explanation.
- Authors wrote “supernatant after centrifugation (7000 rcf, 5 min, good separation)” but what means good separation? Some scale was applied?
- Figure 2 caption is incomprehensible.
- 3.1.4 paragraph, One-step synthesis, seems like the compilation of material and methods with results description. If the authors decided for connecting of results with the discussion it should be added some references to compare with earlier reported work.
- Authors wrote “It must be noted that not all hydroxy groups of the anhydroglucose unit (AGU) of cellulose have the same reactivity toward esterification” However below authors wrote that group at the 6 position reacts up to ten times faster ?
Please correct the phrase for reduce of misunderstanding.
- The DoS parameter, its description and calculation details should be moved to the material and methods section.
- The scale bars on figure 5 are illegible.
- The legend on figure 6 seems to be incomplete.
Reviewer 2 Report
Attached separately.

Round 2
Reviewer 1 Report
Dear authors, the article was significantly improved and is suitable to be published in Polymers.